# Self-Reported Non-Celiac Gluten Sensitivity in Brazil: Translation, Cultural Adaptation, and Validation of Italian Questionnaire

**DOI:** 10.3390/nu11040781

**Published:** 2019-04-04

**Authors:** Yanna A. Gadelha de Mattos, Renata Puppin Zandonadi, Lenora Gandolfi, Riccardo Pratesi, Eduardo Yoshio Nakano, Claudia B. Pratesi

**Affiliations:** 1Interdisciplinary Laboratory of Biosciences and Celiac Disease Research Center, School of Medicine, University of Brasilia, Brasilia 70910-900, Brazil; yannagadelha@gmail.com (Y.A.G.d.M.); lenoragandolfi1@gmail.com (L.G.); pratesiunb@gmail.com (R.P.); 2Department of Nutrition, School of Health Sciences, University of Brasilia, Brasilia 70910-900, Brazil; renatapz@yahoo.com.br; 3Department of Statistics, University of Brasilia, Brasilia 70910-900, Brazil; eynakano@gmail.com

**Keywords:** Non-celiac gluten sensitivity, national survey, questionnaire validation.

## Abstract

This study aimed to translate, culturally adapt, validate, and apply a questionnaire to the Brazilian non-celiac gluten sensitive (NCGS) population. We also aimed to estimate the prevalence of symptoms which affect Brazilian NCGS. The Brazilian Portuguese version of the NCGS questionnaire was developed according to revised international guidelines. Five-hundred-and-fourty-three participants responded the NCGS questionnaire. We evaluated the reproducibility and validity of the questionnaire which presents valid measures of reproducibility. This is the first specific self-reported validated questionnaire for NCGS patients in Brazilian Portuguese, and the first nationwide characterization of self-reported NCGS in Brazilian adults. Most respondents were female (92.3%), and the main intestinal symptoms reported were bloating and abdominal pain. The most frequent extraintestinal symptoms were lack of wellbeing, tiredness, and depression. We expect that the present study will provide a picture of Brazilian individuals with suspected NCGS, which could help health professionals and governmental institutions in developing effective strategies to improve the treatment and diagnosis of Brazilian NCGS.

## 1. Introduction

Non-celiac gluten sensitivity (NCGS) is a gluten-related disorder, characterized by intestinal and extraintestinal symptoms related to the ingestion of gluten-containing food, in subjects that are not affected by celiac disease (CD) or wheat allergy (WA) [1,2,3]. Currently, due to the lack of specific biomarkers to diagnose NCGS, its diagnosis involves the elimination of CD and WA, followed by a gluten-free diet (GFD), and then a challenge with gluten-containing food. The exclusion is followed by assessing the reduction/remission of symptoms after a strict adherence to a GFD, and observing if symptoms return or worsen with gluten consumption [2,4,5]. Therefore, the diagnostic criteria for NCGS should include self-reported gluten intolerance, negative CD serology and no WA [1,2,6].

Over the years, the number of studies on this topic and the number of patients diagnosed with NCGS has significantly increased. It is difficult to access the exact prevalence of NCGS since we still do not have validated biomarkers for the diagnosis of NCGS [2,7,8,9]. However, previous studies have reported a prevalence rate of NCGS between 0.5 and 13% [1,6,10,11]. Researchers have been attempting to characterize the NCGS rates in different countries [2,8,12,13,14,15,16,17,18]. In Latin America, there are few data about the prevalence or characterization of NCGS [15,19,20,21,22].

In Brazil, only one study evaluated NCGS [19], in which the authors aimed to differentiate CD and NCGS symptoms in 80 Brazilian patients. However, there is no study evaluating NCGS in the Brazilian population nationwide due to the lack of a valid instrument in Brazilian Portuguese. Therefore, the primary purpose of this study was to translate, culturally adapt, validate, and apply a questionnaire to the Brazilian NCGS population. We also aimed to estimate the prevalence of symptomatic adverse reactions to gluten ingestion in Brazilian NCGS subjects. We expect that the present study could provide a picture of Brazilian individuals with suspected NCGS and potentially, this could also help institutions in developing effective strategies to improve the treatment and diagnosis of NCGS.

## 2. Materials and Methods 

This study was approved by the Research Ethics Committee of the University of Brasilia (CEP UNB 2.918.449) and followed the guidelines established by the Declaration of Helsinki. Informed consent was obtained from each patient participating in the study. The study was developed in five steps: (i) translation, (ii) cultural adaptation, (iii) validation of the questionnaire, (iv) evaluation of questionnaire´s reproducibility, and (v) application of the questionnaire to Brazilian self-reported NCGS patients. 

### 2.1. Questionnaire 

This study followed the original version of the questionnaire proposed by Volta et al. [8] and was devised by the Italian Celiac Disease Association and the Italian Celiac Foundation. The original questionnaire consists of 60 questions, five sociodemographic characteristics (gender, age, the region where the patient lives, the name of the center affiliation, and the name of the investigator in charge of the center). Eleven questions related to the presence/absence of laboratory test data regarding CD and WA diagnosis. Six questions regarded who was the first to suspect the individual had NCGS. The other questions evaluated the presence or absence of physical manifestations (diarrhea and/or constipation, abdominal pain, bloating, aphthous stomatitis, nausea, epigastric pain, and reflux) and systemic manifestations (tiredness, ‘foggy mind’, headache, joint, or muscle pain; leg or arm numbness; dermatitis; depression; anxiety; and anemia).

### 2.2. Translation, Cultural Adaptation, and Validation

#### 2.2.1. Translation and Retranslation 

In the translation phase, two bilingual health professionals independently translated the questionnaire from English to Portuguese, emphasizing conceptual rather than literal translation. The English questionnaire was translated into a Brazilian Portuguese 7th-grade reading level to obtain a better understanding of the questions by the general population. After the first translation, both translators, along with two health professionals with extensive experience with gluten-related disorders (GRD), met to resolve any discrepancies and integrate both translations into a single version. The single text was retranslated from Brazilian Portuguese to English, by two different bilingual translators working independently from each other, to confirm its accuracy to the original questionnaire. Lastly, the four translators jointly checked the final questionnaire version for accuracy. An adapted and modified version of the Delphi method [23] was used for the validation process.

#### 2.2.2. Cultural Adaptation, Semantic Evaluation, and Validation 

The validation of an instrument consists of a methodological procedure to evaluate its quality, which is related to the capacity of the tool to accurately measure what it is intended to measure [16]. Therefore, the validation of the questionnaire occurred by the cultural adaptation, semantic evaluation, and content validation analyzed by a panel of experts composed of professionals and researchers recognized in their areas. The expert panel consensus helped in the revision of the instrument and ensured its readability and comprehension [17,18]. 

Twelve health professionals were contacted by email and invited to participate and assist with the cultural adaptation and semantic evaluation of the questionnaire. After obtaining their consent, participants received an email with a link to the questionnaire in Brazilian Portuguese that was placed on SurveyMonkey®—an online survey platform. The online survey contained all 46 questions translated to Brazilian Portuguese. The judges rated the items on a five-point Likert Scale for clarity and, when applicable, made suggestions to improve the questionnaire regarding cultural adaptation, comprehension, and clarity [24,25].

The mean grade for the evaluation of clarity and content validation of each item and semantic evaluation was calculated considering the answers provided by the experts. The degree of agreement among the experts for the assessment of importance and clarity of the items was evaluated through the Kendall (W) coefficient of concordance, which ranges from 0 to 1. High W-values (W ≥0.66) indicate that the experts applied the same standards of evaluation as opposed to Low W-values, which suggest disagreement among the experts [16]. The criteria established for the approval of the item was a minimum of 80% agreement between the experts (W-values ≥0.8) [12]. Items considered unclear were rewritten and subject to further evaluation by the experts. Once the experts approved all questions, two bilingual translators met and compared the new Brazilian Portuguese version of the questionnaire to the original version in English. This phase ensured that the Brazilian Portuguese version of the questionnaire was of appropriate cultural relevance while maintaining its fidelity to the original version.

#### 2.2.3. Reproducibility Analysis 

The reproducibility of the questionnaire was evaluated using ten NCGS patients’ responses. The patients answered the questionnaire, and one week later they were invited to answer the same questionnaire. The test–retest reliability (reproducibility) of the questionnaire was verified by the percentage of absolute agreement and Cohen´s Kappa coefficient. 

### 2.3. Brazilian Questionnaire Application

The final step was to place the NCGS Brazilian Portuguese questionnaire on the SurveyMonkey® platform and apply it to a representative number of Brazilian NCGS patients. The first page of the survey presented the consent form which included the established exclusion/inclusion criteria; where non-celiac (NC) participants had to be 18 years of age or older and have symptoms associated with the ingestion of gluten. At that point, participants gave their consent. Individuals that did not agree to participate were directed to a page thanking them for their time; while those that agreed were directed to the first page of the survey containing six social demographic questions. The final part of the study consisted of applying the 40 translated and culturally adapted questions to Brazilian NCGS patients.

## 3. Results

### 3.1. Translation, Cultural Adaptation, Semantic Evaluation, and Content Validation

The summary of the stages of the Brazilian questionnaire process is displayed in Figure 1. The questionnaire was constructed considering the translation/retranslation and suggestions made by the experts. After the translation/retranslation steps, the content validation and the semantic evaluation was performed by experts, who decided which questions to retain [7]. Some questions in the original questionnaire related to the presence/absence of laboratory results that were eliminated. The original questionnaire was applied and answered in one of the 38 celiac clinics in Italy, where both respondents and physicians had access to patient records; the Brazilian version is a self-response, online questionnaire, and therefore called for a simplified version. 

We also grouped the six questions about who was the first to suspect NCGS (Appendix A, question 44). Regarding the patient’s characterization, we maintained the questions about gender, age, and where they lived, and included questions about income, marital status, and educational level (Appendix A, questions 1 to 6). Therefore, the Brazilian questionnaire final version comprised 46 questions (Appendix A).
A total of three rounds were necessary to obtain an agreement among the experts for content validation and semantic evaluation. In the first round of the questionnaire, questions were considered adequate regarding reliability, clarity and easy comprehension. The questionnaire was sent to 17 experts, and only nine completed the evaluation in the first round. In the first round, almost 92% (N = 55) of the questions were approved. The experts suggested changes in the questions that were not approved and only those questions were sent in the second round for evaluation. In the second round, we obtained answers from 11 experts, and 89% of the questions were approved. Therefore, the third round of expert evaluation was necessary for the remainning questions. The experts suggested grouping the questions as previously described, and all questions were approved in the third round.

#### Reproducibility of the Brazilian NCGS Questionnaire

The reproducibility of the questionnaire was verified by Cohen’s kappa coefficient and the percentage of absolute agreement. All items relating to the gastrointestinal symptoms, extraintestinal manifestation, and disorders associated to NCGS presented statistically significant kappa values (*p* < 0.05) and absolute agreement equal to or greater than 80%, indicating good reproducibility of the questionnaire.

### 3.2. Brazilian NCGS Questionnaire Application

The questionnaire was disseminated through email and social-media nationwide to adults in Brazil who suspected NCGS. In total, 561 individuals answered the questionnaire. However, 18 individuals were excluded because they did not meet the inclusion criteria. Therefore, our sample was composed of 543 adult individuals (92.3% female, N = 501; mean age 38.2 ± 9.5 years) from all of five Brazilian regions. Most patients reported more than two associated gastrointestinal or extraintestinal symptoms (Figure 2 and Figure 3). Regarding gastrointestinal symptoms, the most frequent were bloating (93%), abdominal pain (74.3%), and heartburn (71.6%). Among the Brazilian NCGS suspects, constipation was more frequently reported (66.7%) than diarrhea (45.1%) (Figure 2). Regarding family history, 11% (N = 60) of patients with suspected NCGS had a first- or second-degree relative affected by CD. 

The most frequent extraintestinal manifestations were lack of wellbeing (89.5%), tiredness (77%), and depression (70.2%) (Figure 3). Also, a high prevalence of neuropsychiatric symptoms was mentioned, including headache (63.9%), ’foggy mind’ (59.9%), and anxiety (52.1%). Other extraintestinal manifestations were joint (49.7%) and muscle pain (45.9%), skin rash (39.4%), rhinitis (37%), numbness (33%), anemia (22.7%), and weight loss (15.7%). Less than 10% of patients disclosed symptoms related to asthma.

The most frequent disorder in patients with NCGS was food intolerance (64.8%), followed by irritable bowel syndrome (IBS) (46.4%), and allergies (nonskin: 46.2%; skin: 41.8%) (Figure 4). Approximately 36% of patients presented psychiatric disorders. One or more associated autoimmune diseases were present in 20.6% of patients. Eating behavior disorders preceded the suspected diagnosis of NCGS in 6.3% of cases.

In more than 50% of our cases NCGS was suspected by the patients (Table 1). Only in 15.8% of the cases the gastroenterologist suspected of the existence of NCGS, followed by the patient’s general practitioner (4.8%%), friends (4.2%), homeopath practitioner (1.5%), or pharmacist (0.5%).

## 4. Discussion

Unlike CD, which was first described over 8000 years ago [26], the first studies on NCGS were published in the late 1970s and early 1980s [27]. However, since 2010 the number of studies on NCGS has grown as have the sales of gluten-free food (GFF), and both are expected to continue growing in the coming years [1,2]. The absence of biomarkers makes it difficult to diagnosis NCGS. Consequently, the diagnosis of NCGS is based on clinical response to gluten ingestion and withdrawal, followed by gluten challenge after both CD and WA have been excluded [2]. Therefore, studies of symptoms and associated disorders are critical to provide a better approach to diagnosis. To the best of our knowledge, our study is the first nationwide characterization of NCGS Brazilian adults. We validated the first specific self-reported questionnaire for NCGS patients in Brazilian Portuguese, based on the Volta et al. [8] questionnaire. 

We followed the recommended linguistic validation process (translation and retranslation) because the original instrument was in a language other than the target language, and there was no translated and validated version [28]. Therefore, the first step of this study was to translate/retranslate the original version of the questionnaire to English/Portuguese/English. The cross-cultural adaptation process followed the guidelines predominant in the literature after the translation process [29,30]. We measured the comprehension of the instrument through the semantic evaluation. This step ensures that the instrument is easily understood and clear [31]. The Brazilian Portuguese versions of the NCGS questionnaire demonstrate cultural and semantic adequacy, and therefore represent the first Brazilian Portuguese version developed. According to the reproducibility evaluation, the questionnaire presented proper measures of reproducibility, which indicates that similar results under consistent conditions are reproducible. 

After the validation, the instrument was sent nationwide to Brazilian NCGS subjects to evaluate their characteristics. The respondents were predominantly female (92.3%) and experience both intestinal and extraintestinal symptoms on gluten ingestion similarly to participants in other studies [8,10,14,20]. The high percentage of female respondents was expected since women tend to be more concerned about health, and more frequently participate in health studies [32,33,34,35]. Similarly, a survey conducted in the United Kingdom [10], which evaluated the population prevalence of self-reported gluten sensitivity, showed that almost 80% of the individuals who self-reported NCGS were female. Another study [14], conducted in the Netherlands, also showed a higher prevalence of NCGS among women (60%) than men. Our results corroborate these previous studies. 

According to the Italian prospective multicenter study by Volta et al. [8] and a systematic review [1], the main NCGS gastrointestinal symptoms are bloating (87%), abdominal pain (83%), diarrhea (>50%), alternating bowel habits (27%), constipation (24%), and epigastric pain (52%). Nausea, acid reflux, aerophagia, and aphthous stomatitis were the symptoms less frequent. In our study, the most frequent gastrointestinal symptoms were bloating (93%), abdominal pain (74.3%)—similarly to the Italian study—and heartburn (71.6%). In contrast to the Italian study [8], the Brazilian NCGS subjects presented a higher percentage of constipation (63.0%), and the constipation prevalence was higher than the diarrhea prevalence (45.1%). In our study, almost 65% of the patients reported alternating bowel habits. The participants in the UK study by Aziz et al. [10] had a similar age range and intestinal symptoms, such as bloating (78%), abdominal discomfort/pain (67%), and altered bowel habit (37%), which was similar to our results. In a study conducted in Argentina, the most common gastrointestinal symptoms reported by NCGS patients were bloating (70.1%), and abdominal discomfort (47.1%) [20]. A study conducted in the Netherlands [14] reported that the most frequent intestinal symptoms in NCGS were bloating (almost 80%) and abdominal discomfort (more than 50%), which is similar to the study conducted in Mexico [21], in which the most frequent intestinal symptoms were bloating (81%) and abdominal discomfort (42%). The Mexican study showed that these symptoms were mitigated by following a GFD for four weeks (bloating: 25%; abdominal discomfort: 14%) in NCGS patients (N = 12) [21].

Among extraintestinal symptoms, the Italian study [8] showed that tiredness (64%) and lack of well-being (68%) are common between NCGS patients, followed by foggy mind (38%), headache (54%), muscle or joint pain (31%), arm/leg numbness (32%), anxiety (39%), or depression (18 %). Participants in the Italian study also reported weight loss (25%), dermatitis (18%) or skin rash (29%), and anemia (22%) [8]. In Brazil, the most frequent extraintestinal manifestations were lack of wellbeing, tiredness, and depression. All of these manifestations had a higher prevalence (Figure 3) than the Italian study. Brazilians’ NCGS presented higher prevalence than Italians of neuropsychiatric symptoms including headache (63.9%), foggy mind (59.9%), and anxiety (52.1%). The only manifestation that was higher in Italians than in Brazilians was weight loss (Figure 3). In the study conducted in the Netherlands [14], tiredness (almost 40%) and headache (almost 20%) were the most frequent extraintestinal symptoms reported by NCGS patients. Similar to the Netherland study, in the UK study [10] the most frequent extraintestinal symptoms were tiredness (23%) and headaches (22%), lower than in Brazilian and Italian [8] subjects. Extraintestinal symptoms such as anxiety and depression are common among patients with food hypersensitivity [36]. In people who suffer from gluten-related disorders, these symptoms are frequently associated as a consequence of food restriction, and not as a symptom caused by gluten ingestion [10]. It is noteworthy that our results are similar to the literature in which the NCGS symptoms commonly found include bloating, abdominal pain, diarrhea, epigastric pain, nausea, aerophagia, lack of well-being, tiredness, headache, foggy mind, and anxiety [5,8,10,14,37].

In our study the most frequent associated disorder was food intolerance (64.8%) to other foods or ingredients. The association of one or more food restrictions can have a detrimental effect on the quality of life of the NCGS patient and impact the treatment [24,38] since GFD can lead to social restriction and therefore have a negative emotional impact [24,39,40]. In our study, ~70% of NCGS patients mentioned depression that can be associated with NCGS and or GFD. A systematic review with meta-analysis regarding mood disorders and gluten [40] showed that gluten restriction represents an effective treatment strategy for mood disorders in individuals with gluten-related disorders, including NCGS. Therefore, the high percentage of individuals with NCGS who suffer from mood disorders could benefit from a GFD.

Regarding family history, 11% (N = 60) of patients with suspected NCGS had a first- or second-degree relative affected by CD. Similarly, the study conducted in the UK [10] showed a prevalence of 12.4% of individuals with a relative affected by CD. The Italian study [8] found that 12.8% of NCGS patients had a family history of CD, which is similar to our findings. 

Symptoms such as, fatigue, abdominal pain, alternating bowel habits, and bloating are common in NCGS, CD, and IBS [41,42]. Of those that responded the NCGS, 46.4% reported suffering from IBS. A review by Usai-Satta et al. [37] showed that IBS and NCGS share symptoms such as diarrhea, constipation, or abdominal pain. Therefore IBS is often considered as part of the NCGS. However, the study emphasizes that for subjects diagnosed with NCGS, the ingestion of gluten exerts a direct effect on the onset of digestive symptoms, and only a portion of the patients with IBS relate their symptoms to a gluten-containing diet [37]. Therefore, the presence of IBS in NCGS patients appears to be common, but IBS is not common among NCGS patients [41,42].

There can be a discrepancy between the perceived NCGS and the gold standard method of testing, which is comprised of dietary elimination of gluten followed by double-blind, randomized, placebo-controlled food challenges. Also, it is uncertain whether it is gluten withdrawal or withdrawal of another component present in wheat (or other gluten-containing foods) that benefits patients; because this is an online questionnaire, we are not able to investigate further. 

## 5. Conclusions

In this study, a cross-cultural Brazilian Portuguese version of the NCGS questionnaire was translated, culturally adapted, validated, and applied to a self-reported Brazilian NCGS population. It is, to the best of our knowledge, the first nationwide characterization of NCGS Brazilian adults. We estimated the prevalence of symptoms that affect the target population and described the frequency of both intestinal and extraintestinal signs and symptoms of NCGS in Brazilian subjects. In our study most of the participants were females, and the main intestinal symptoms were bloating, and abdominal pain; the most frequent extraintestinal symptoms observed were lack of wellbeing, tiredness, and depression. Our results do not differ much from what was found in similar studies performed in other countries. 

Although the terms gluten and gluten sensitivity have become commonplace, disorders associated with gluten ingestion remain poorly understood. Patients suffering from NCGS are probably a heterogeneous group, composed of several subgroups each characterized by different pathogenesis, clinical history, and clinical course [43]. The only event common to all individuals suffering from NCGS is the appearance of a varied range of adverse signs and symptoms after ingestion of gluten. Future research is needed to identify reliable biomarkers for NCGS diagnosis, which would allow a better definition of each NCGS subgroup.

## Figures and Tables

**Figure 1 nutrients-11-00781-f001:**
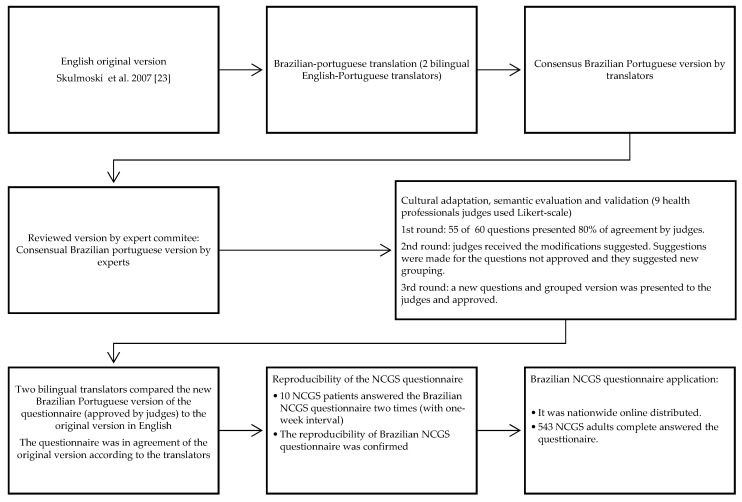
Sequential stages followed in the process of translation, cultural adaptation, and validation of the Brazilian non-celiac gluten sensitive (NCGS) questionnaire.

**Figure 2 nutrients-11-00781-f002:**
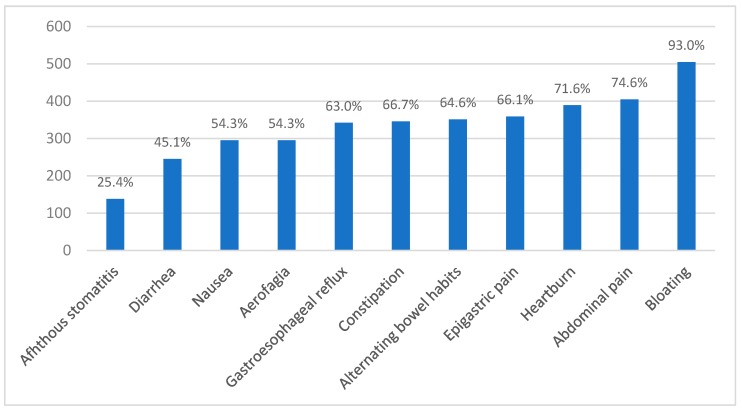
Gastrointestinal symptoms in Brazilian NCGS subjects (N = 543).

**Figure 3 nutrients-11-00781-f003:**
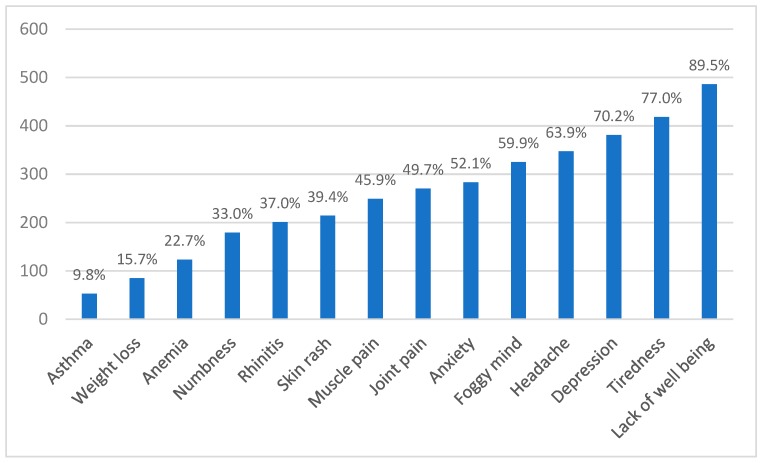
Extraintestinal manifestations in Brazilian with suspicion of non-celiac gluten sensitivity (N = 543).

**Figure 4 nutrients-11-00781-f004:**
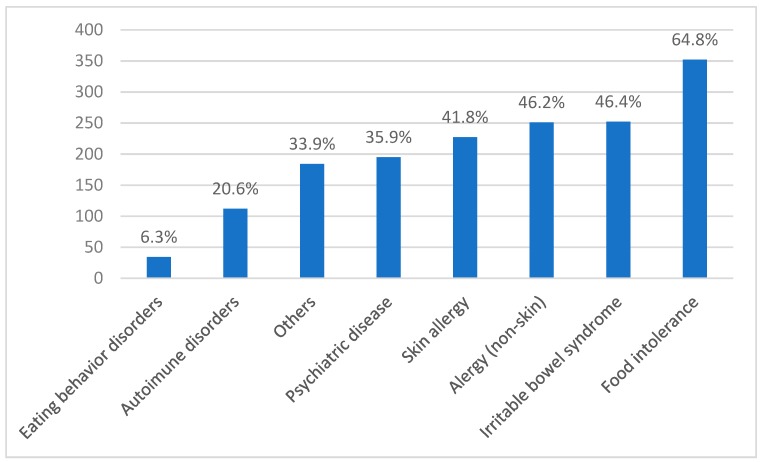
Disorders associated with Brazilian suspected non-celiac gluten sensitivity (N = 543).

**Table 1 nutrients-11-00781-t001:** Individuals who initially suspected the possible existence of NCGS: frequency and prevalence of the 543 individuals.

	Frequency (N)	Prevalence (%)
Pharmacist	2	0.4%
Homeopath	8	1.5%
Friends	23	4.2%
General Practitioner	26	4.8%
Gastroenterologist	86	15.8%
Others	116	21.4%
Patient	282	51.9%

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
