# Peer review of "Self-Reported Non-Celiac Gluten Sensitivity in Brazil: Translation, Cultural Adaptation, and Validation of Italian Questionnaire"

_nutrients, 2019, doi:10.3390/nu11040781_

Reviewer 1 Report

This is interesting manuscript.

The authors describe in detail how they prepared questionnaire from Italian country to Brazilin Portuguese population. They not only translate the text but also culturally adopt it. This is very important for it suitability in this new population. The number of participants who have completed this questionnaire is small.Therefore for me the greatest value of this manuscript is description how to prepare questionnaire to selected population.

Author Response

Thank you for taking the time to review our study!

Reviewer 2 Report

You chose to report only the frequency of reported symptoms. Did you consider looking at the underlying dimensions of the data for specific combinations of symptoms? Exploratory factor analysis would identify symptom groups and reveal the underlying dimensionality of symptoms which could then be compared to Rome IV criteria. 

The sentence at line 131 was unclear. Did you mean to say "applied the original questionnaire which respondents answered at one of the 38 celiac clinics in Italy where both respondents and physicians had access to patient records.." ?

Line 289 has mixed tenses. As this was in the past, the use of translated, validated and applied is indicated. 

Author Response

In response to your first question:

Our main objective was to translate and validate the questionnaire. Now that we have a validated questionnaire we can, apply it in a more controlled setting,  further our study. 

In response to observations 2 and 3 - 

I made changes to the text. 

 Thank you for taking the time to review our study.

Claudia